# Comparing Gene Expression in the Parabrachial and Amygdala of Diestrus and Proestrus Female Rats after Orofacial Varicella Zoster Injection

**DOI:** 10.3390/ijms21165749

**Published:** 2020-08-11

**Authors:** Rebecca Hornung, Addison Pritchard, Paul R. Kinchington, Phillip R. Kramer

**Affiliations:** 1Department of Biological Sciences, Texas A&M University College of Dentistry, Dallas, TX 75246, USA; hornung@tamu.edu (R.H.); apritchard@tamu.edu (A.P.); 2Departments of Ophthalmology and Molecular Microbiology and Genetics, University of Pittsburgh, 203 Lothrop Street, Pittsburgh, PA 15213, USA; kinchingtonp@upmc.edu

**Keywords:** orofacial, herpes zoster, shingles, pain, steroids

## Abstract

**** The orofacial pain pathway projects to the parabrachial and amygdala, and sex steroids have been shown to affect neuronal activity in these regions. GABA positive cells in the amygdala are influenced by sex steroid metabolites to affect pain, and sex steroids have been shown to alter the expression of genes in the parabrachial, changing neuronal excitability. Mechanisms by which sex steroids affect amygdala and parabrachial signaling are unclear. The expression of genes in the parabrachial and amygdala in diestrus (low estradiol) and proestrus (high estradiol) female rats were evaluated in this study. First, varicella zoster virus was injected into the whisker pad of female rats to induce a pain response. Second, gene expression was quantitated using RNA-seq one week after injection. Genes that had the greatest change in expression and known to function in pain signaling were selected for the quantitation of protein content. Protein expression of four genes in the parabrachial and seven genes in the amygdala were quantitated by ELISA. In the parabrachial, neurexin 3 (Nrnx3) was elevated at proestrus. Nrnx3 has a role in AMPA receptor and GABA signaling. Neuronatin (Nnat) and protein phosphatase, Mg^2+^/Mn^2+^ dependent 1E (Ppm1e) were elevated in the parabrachial of diestrus animals both genes having a role in pain signaling. Epoxide hydroxylase (Ephx2) was elevated in the parabrachial at proestrus and the vitamin D receptor (Vdr) was elevated in the amygdala. Ephx2 antagonists and vitamin D have been used to treat neuropathic pain. In conclusion, sex steroids regulate genes in the parabrachial and amygdala that might result in the greater pain response observed during diestrus.

## 1. Introduction

Orofacial pain is influenced by sex steroids [1,2,3]. Sex steroids alter gene expression in the trigeminal ganglia, trigeminal nucleus, and thalamus to regulate orofacial pain [4,5,6,7,8]. The affective/motivational pain pathway from the trigeminal region projects to the parabrachial and amygdala, with few direct projections to the amygdala [9,10,11]. Interestingly, GABA positive cells in the amygdala are influenced by sex steroid metabolites to affect pain [12]. Also, the intra-amygdala application of sex steroids alter the pain response [13]. The parabrachial controls orofacial pain signaling [14], and sex steroids have been shown to alter the expression of genes in the parabrachial [15]. These sex steroids then change the excitability of neurons within this region [16]. The estrogen receptors are expressed in the parabrachial and amygdala region and could signal the sex steroid response [17,18]. Currently, the mechanisms of alterations in gene expression in the amygdala and parabrachial by sex steroids are not clear.

Varicella zoster virus (VZV) infection results in viral latency [19,20,21,22], and often, VZV re-activates resulting in what is termed herpes zoster (HZ). Unfortunately, HZ causes pain in up to 90% patients [23]. This zoster-associated pain occurs before or near the time skin lesions are observed, and post-herpetic neuralgia (PHN) is pain defined as beginning 3 months after the resolution of the lesions [24,25,26]. VZV injected into the footpad of either Wistar or Sprague Dawley rats results in measurable hypersensitivity that is similar to that in humans with PHN [27,28,29,30,31,32,33]. The treatment of these animals with antivirals, opioids, or NSAIDS mirrors PHN patients treated with these same drugs, in that zoster-associated pain does not respond to antivirals, but does respond to treatment with gabapentin or sodium channel blockers [29,30,33,34]. In models of post-herpetic neuralgia, sex steroids alter the orofacial pain response [35]. Interestingly, females report PHN 3.75 times more often than men such that this pain has been reported to be 36% higher in females [36,37]. Thus, altered gene expression due to changes in sex steroids would be expected in this model.

Because the parabrachial and amygdala are important affective pain pathways [38], gene expression in the parabrachial and amygdala was compared in diestrus (low estradiol) and proestrus (high estradiol) rats. One week after VZV injection of the whisker pad, the parabrachial and central amygdala tissue were isolated and the gene expression was quantitated using RNA-seq. Gene expression between diestrus and proestrus animals were compared in VZV and control animals. Genes having the greatest change in expression and known to function in pain signaling were selected for the quantitation of protein content in the parabrachial and amygdala tissue by ELISA.

## 2. Results

The analysis of the gene expression was compared between female rats in diestrus and proestrus. Estrous cycle determination was based on vaginal smears and the concentration of estradiol in the plasma (Figure 1 and Table 1). Significant changes in gene transcript were identified for the parabrachial region (Table 2 and Appendix A) and the central amygdala (Table 3 and Appendix A) after VZV injection of the left whisker pad. Rat 3 in the VZV treated group was not analyzed because the estradiol levels, as measured by ELISA, were not distinctive of diestrus (Table 1).

Genes to be analyzed at the protein level were selected on two criteria: first, the genes that had the greatest change in fpkm between diestrus and proestrus were analyzed (*p* value of < 0.005), and second, the gene had been reported to function in pain. In these studies, we report changes in gene expression after injecting VZV. Neurexin 3 (Nrnx3) increased at the transcript level (*p* = 0.003) (Figure 2A) and at the protein level (Figure 2B,C). The protein content for Nrnx3 in the diestrus/contralateral group (Mdn = 1.18) was lower than proestrus/contralateral group (Mdn = 1.46) and this difference was significant U(*n* diestrus = 6, *n* proestrus = 6) = 5, *p* = 0.04. Combining the data from the right and left parabrachial the proestrus Nrnx3 expression was greater at proestrus (Mdn = 1.39) versus diestrus (Mdn = 1.18) and this difference was significant U (*n* diestrus = 12, *n* proestrus = 12) = 5, *p* = 0.01.

Expression of neuronatin (Nnat) increased at diestrus after VZV injection, when comparing the diestrus/VZV group to the proestrus/VZV group (Figure 3A). This change was also observed in protein content (Figure 3B), compare the diestrus/contralateral group (Mdn = 12.60) to the proestrus/contralateral group (Mdn = 9.28), this difference was significant U(*n* diestrus = 6, *n* proestrus = 6) = 0, *p* = 0.002. Combining the data from the right and left parabrachial there was no significant change in Nnat expression (Figure 3C).

Epoxide hydrolase 2 (Ephx2) transcript significantly increased (*p* value = 5.5 × 10^−21^, Table 2) in the parabrachial nucleus of proestrus animals injected with VZV (Figure 4A). This change at the transcript level was also observed in the amount of protein (Figure 4B). When comparing the diestrus/contralateral group (Mdn = 0.77) to the proestrus/contralateral group (Mdn = 1.13), this difference was significant U(*n* diestrus = 6, *n* proestrus = 6) = 1, *p* = 0.004. Combining the right and left side (Figure 4C) diestrus animals (Mdn = 0.77) had lower expression than proestrus animals (Mdn = 1.05) and this difference was significant U(*n* diestrus = 12, *n* proestrus = 12) = 15, *p* = 0.0005.

Protein phosphatase, Mg^2+^/Mn^2+^ dependent 1E (Ppm1e) transcript was higher in the parabrachial nucleus of diestrus animals versus proestrus animals after VZV injection (Figure 5A). A change in the amount of protein was also observed (Figure 5B), comparing the diestrus/ipsilateral group (Mdn = 5.19) versus proestrus/ipsilateral group (Mdn = 2.35). This difference was significant U(*n* diestrus = 6, *n* proestrus = 6) = 0, *p* = 0.002. The diestrus/ipsilateral group (Mdn = 5.19) was also greater than proestrus/contralateral group (Mdn = 3.34) and this difference was significant U(*n* diestrus = 6, *n* proestrus = 6) = 0, *p* = 0.002. When combining the right and left side parabrachial tissue there was no significant difference between the diestrus and proestrus animals (Figure 5C).

After VZV injection the vitamin D receptor (Vdr) transcript was higher in the central amygdala of proestrus rats versus diestrus rats (Table 3 and Figure 6A). Vdr protein levels were significantly greater in the proestrus/ipsilateral group (Mdn = 0.75) versus diestrus/ipsilateral group (Mdn = 0.40), U(*n* diestrus = 6, *n* proestrus = 6) = 5, *p* = 0.041. On the contralateral side the proestrus/contralateral group (Mdn = 0.67) had a greater amount of protein compared to the diestrus/contralateral group (Mdn = 0.33) and this difference was significant U(*n* diestrus = 6, *n* proestrus = 6) = 3, *p* = 0.015. When combining the right and left side amygdala samples, the proestrus animals showed a significantly greater amount of Vdr protein than diestrus animals (Figure 6C); in the proestrus group (Mdn = 0.68) there was a greater amount of protein as compared to the diestrus group (Mdn = 0.36) and this difference was significant U(*n* diestrus = 12, *n* proestrus = 12) = 18, *p* = 0.001.

Significant changes in transcript of the central amygdala were observed when comparing diestrus and proestrus animals (Table 3 and Figure 7), but no significant changes were detected in protein expression (data not shown).

## 3. Discussion

Previous studies demonstrate a significant difference in the motivational/affective pain response between proestrus and diestrus rats [40]. Brain regions involved in regulating affective pain responses include the parabrachial and amygdala [10]. In a screen for changes in transcript expression between different phases of the estrous cycle, multiple genes in both the parabrachial and amygdala were differentially expressed in proestrus and diestrum rats. Many of these genes have been shown to function in pain signaling. Pain was induced in these animals by injection of VZV in the whisker pad. A saline control was performed to contrast changes in gene expression due to MeWo cell and virus injection although MeWo cells without virus are the typical control to determine the effects of VZV [27,30,33]. Further studies were completed on genes that had the greatest change in transcript between proestrus and diestrus and on genes shown to have a role in pain processing. The protein expression for four genes was quantitated in the parabrachial and for seven genes were quantitated in the amygdala.

Nrxn3 expression was greatest in the parabrachial during proestrus, the period during which the sex steroids are highest. The deletion of Nrxn3 decreases the amplitude of excitatory potentials, possibly through decreasing AMPA receptor levels [41]. Interestingly, Nrxn3 can also decrease GABAergic inhibitory responses [41]. Because we observed a decrease in the pain response at proestrus [40], an increase in Nrxn 3 at proestrus might function by decreasing the excitatory response in the parabrachial region. A closer look at AMPA receptor levels on the membrane of the neurons could suggest that Nrxn3 affects the pain response by decreasing surface AMPA receptor level during proestrus.

Nnat is present in the dendrites of neurons and has a role in synaptic plasticity [42]. Nnat alters calcium levels to affect neuronal excitability, Nnat overexpression results in elevated cytoplasmic and dendritic calcium levels, potentially enhancing excitability of the neuron [42]. Nnat has also been correlated with neuropathic pain [43]. Elevated levels of Nnat at diestrus could result in greater excitability and an increased pain response after VZV injection [40].

Ephx2 is a pharmaceutical target for control of inflammatory pain [44]. This soluble epoxide hydroxylase converts anti-inflammatory eicosanoids to a less active diol form [44]. High levels of Ephx2 would be expected to decrease the level of these anti-inflammatory lipids. In this study, Ephx2 increased at proestrus and would be expected to decrease the anti-inflammatory lipids. This is counter to our results that proestrus showed the lowest pain response when Ephx2 levels were highest.

A fourth gene was Ppm1e, expression was elevated at diestrus but only on the ipsilateral side of the parabrachial. Ppm1e contributes to dephosphorylation of 5′-AMP-activated protein kinase (AMPK), deactivating AMPK [45,46]. AMPK activation reduces pain [47,48], thus, an increase in Ppm1e would reduce AMPK activation resulting in an increased pain response. We observed an increased pain response during diestrus [40]. This increased pain response could be due to an increase in Ppm1e at diestrus. However, signaling for the pain would be expected to occur through a contralateral pathway. Because Ppm1e was not elevated on the contralateral side during diestrus, the role of Ppm1e is unclear.

Lastly, the vitamin D receptor was elevated in the central amygdala at proestrus or when sex steroids are highest. The vitamin D receptor has been reported in the amygdala previously [49]. Application of vitamin D has been efficacious in reducing the pain response in rats [50]. It was hypothesized that vitamin D would affect the expression of several opioid genes, leading to pain relief [50]. Elevation of the vitamin D receptor could trigger an increase in opioid signaling due to neurons being more sensitive to vitamin D. If so, the elevation of the vitamin D receptor at proestrus would lead to an increase in opioid signaling within the amygdala and reduce the pain response during proestrus, as previously observed [40]. Of the seven genes quantitated at the protein level in the amygdala only, Vdr was significantly different between the proestrus and diestrus rats. Note that the level of expression for Vdr was manifolds lower than most genes. For example, the Vdr fkpm ranged from 0.01 to 1.3 in the amygdala but Nrxn3 ranged from 40 to 60. When looking at protein expression, Vdr ranged between 0.4 and 0.7 pg/mg but Nnat was between 10 and 13 pg/mg (mean values). Thus, not only were there differences between treatment groups, there were also dramatic differences between the level of transcript between different genes.

Future experiments will test if the changes in expression for the genes described alter the behavioral response and then the mechanism for this change can be dissected. Other genes that had a smaller change in transcript between diestrus and proestrus, that are known to regulate pain, could be examined further. For example, the Solute Carrier Family 30 Member 3 (Slc30a3) gene was elevated in the amygdala during the diestrus phase. Protein expression was not quantitated for Slc30a3 but in the event that the current set of genes do not have a significant role in the pain response, additional genes from the screen can be examined, although the change in expression between the diestrus and proestrus animals was not as great as the genes outlined in this study.

In conclusion, several genes that have a known role in pain signaling showed altered expression in either the parabrachial or amygdala in different phases of the estrous cycle. There is potential that these genes have some role in the pain response observed between diestrus and proestrus rats injected with VZV [40]. Further mechanistic studies will focus on why the expression changes between the different phases of the estrous cycle and if there are promoter elements that allow for sex steroids to regulate the gene’s expression.

## 4. Methods

### 4.1. Animal Welfare

All work with animals was approved by the Texas A&M University College of Dentistry Institutional Animal Care and Use Committee (IACUC 2019-0353-CD). Female Long Evans rats (280 g) were purchased from Envigo (Indianapolis, IN, USA) and kept on a 14:10 light/dark cycle. The rats were given food and water ad libitum. After a 4-day acclimation period, the experiments were carried out in accordance with the NIH regulations on animal use.

### 4.2. Treatment Groups

The expression of all coding RNA (i.e., rat cDNA library) was quantitated by RNA-seq analysis. There were four groups of rats analyzed for RNA transcript expression. Twelve female rats were used in this experiment and half received a unilateral injection of VZV and the other half vehicle. One week later, these two groups were divided and half were sacrificed at diestrus and the other half during proestrus, resulting in four groups: diestrus/control, diestrus/VZV, proestrus/control, and proestrus/VZV. To determine the stage of the estrous cycle in which to collect brain tissue, we utilized both vaginal smear data and the estradiol concentration. Because the concentration of plasma estradiol varies over the estrous cycle (Figure 1A), the differences between diestrus and proestrus should not be the only indicator and the vaginal smears aid in a positive determination (Figure 1B–D).

For the analysis of protein content, 12 animals received a VZV injection in the left whisker pad and then one week later half were sacrificed at diestrus and the other half at proestrus. When the parabrachial or amygdala tissue were isolated, the right and left sides were isolated separately. Brain tissue isolated from the right was contralateral to the VZV injection and brain tissue isolated from the left was ipsilateral to the VZV injection. The groups were divided into 4 further groups, diestrus/ipsilateral, diestrus/contralateral, proestrus/ipsilateral, and proestrus/contralateral with 6 animals per group. If a significant effect was observed due to VZV or estrous cycle the right and left side data was combined in a separate data analysis.

### 4.3. Vaginal Smear Protocol

The vaginal smears were performed daily on female rats to determine the stage of the estrous cycle. Each female rat’s vagina was lavaged twice a day at 0800 and 1500 h using 250 μL of sterile 0.9% saline, and the solution was then transferred to a glass slide (StatLab, Inc., Lewisville, TX, USA). The slides were completely dried and then fixed and stained using a Hema-Diff rapid differential stain kit (Anapath; StatLab, Mckinney, TX, USA). After staining, the cell morphologies were observed with a microscope (Figure 1B–D) and recorded.

### 4.4. VZV Infection and Tissue Punches

The rats were anesthetized with 2% isoflurane and an air flow of 2 L per minute. Each rat’s left whisker pad was injected with 100 µL of MeWo cells infected with VZV (50,000–100,000 pfu/µL) or vehicle (0.9% saline). The viral titers were distributed so that 1–2 animals in each group were injected with the lower titer cells (i.e., 50,000 pfu/µL) and 1–2 animals were injected with the higher titer cells (i.e., 100,000 pfu/µL). One week after VZV infection, fresh parabrachial or central amygdala tissue (5–10 mg) was collected within 1 h of obtaining the vaginal smear. The animals were euthanized by exposure to CO_2_ followed by decapitation. The brain was removed using a rongeur and placed on a brain slicer (Zivic, Pittsburgh, PA, USA). After cooling using dry ice, 1-mm thick sections were cut. For the parabrachial, the slice between Bregma −8.6 to −9.6 was collected. For the central amygdala, the slice between Bregma −1.8 to −2.8 was collected. These sections were placed on glass slides and kept on dry ice. Parabrachial tissue was collected with punches 2 mm in diameter centered on coordinates 2.0 mm lateral of midline and a depth of 7.0 mm from the top of the skull. For the central amygdala, a 2 mm punch was collected centered on coordinates 4.0 mm lateral of midline and a depth of 8.2 mm from the top of the skull [51]. The tissue was stored in liquid nitrogen until use.

### 4.5. RNA-Seq Analysis

Sequencing libraries were generated from 2 μg of purified RNA using NEBNext UltraTM RNA Library Prep Kit for Illumina and index codes were added to attribute sequences to each sample (Novogene Corporation, Beijing, China). Paired end reads (150 bp, average 20 million reads) per sample were obtained using the HiSeq 4000 platform from Illumina. The sequencing reads were aligned to the rat genome using TopHat v2.0.12 [52]. The reads numbers mapped to each gene was counted using HTSeq v0.6.1. Fragments per kilobase of transcript per million fragments mapped (fpkm) of each gene was calculated based on the length of the gene and reads count mapped to this gene.

### 4.6. Enzyme-Linked Immunosorbent Assay (ELISA)

The tissue was placed in 250 μL of T-Per tissue protein extraction reagent containing Halt Protease Inhibitor and ground (Thermo Scientific, Rockford, IL, USA). Ground samples were frozen and thawed, followed by centrifugation and decanting of the supernatant. The quantitation of protein in the tissue samples was performed in duplicate for Nrnx3, Nnat, Ephx2, Ppm1e, Kcnip1, Scn5a, Kcng1, Vdr, and Nptxr using ELISA (MyBioSource, San Diego, CA, USA). Total protein was determined in each sample using a BCA protein assay (Thermo Scientific, Waltham, WA, USA). Values represent the pg of protein measured by ELISA per mg of total protein. Blood was also collected after the animals were euthanized and the estradiol levels were measured in the plasma to confirm the estrous cycle stage as determined from the vaginal smears. To measure estradiol levels, the estradiol ELISA kit was used according to the manufacturer’s directions and samples were analyzed in duplicate (catalog # KGE014, R & D systems, Minneapolis, MN, USA).

### 4.7. Statistical Analysis and Data

ELISA and RNA-seq data was analyzed with the non-parametric Mann–Whitney test (Prizm 5.04, GraphPad Software, La Jolla, CA, USA). The data that support the findings of this study is available from the corresponding author upon reasonable request. Note: RNA-seq data included 2–3 animals per treatment group, power analysis calculations for a level 0.05 tests using the mean and standard deviation values indicated a range of 30% power for the Vdr gene to over 90% power for the Ppm1e and Ephx2 gene. A standard of 80% power is acceptable but we are below that standard for Nrnx3, Nnat, Scn5a, Vdr and Nptxr.

## Figures and Tables

**Figure 1 ijms-21-05749-f001:**
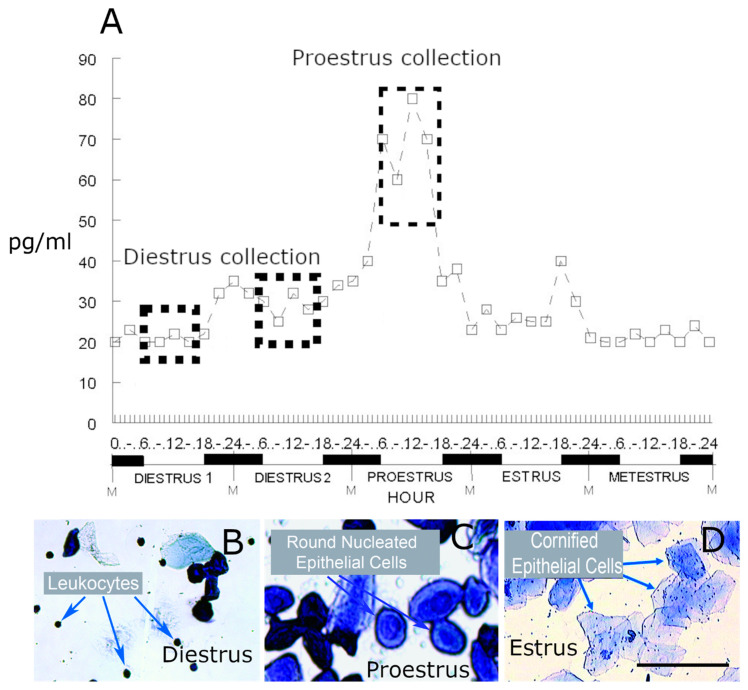
In panel (**A**) the mean concentration values (pg hormone/mL of blood plasma) for 17β-estradiol during the estrous cycle of intact, female rats, used with permission, Butcher et al., 1974 [39]. Vaginal smears during the light phase are shown for diestrus (panel **B**), proestrus (panel **C**) and estrus (panel **D**). The predominant type of cell present during a particular estrous phase is described on the panels (**B**–**D**). Animals were sacrificed during the light phase during diestrus and proestrus (dotted boxed regions in panel **A**). M = Midnight. Bar = 100 µM

**Figure 2 ijms-21-05749-f002:**
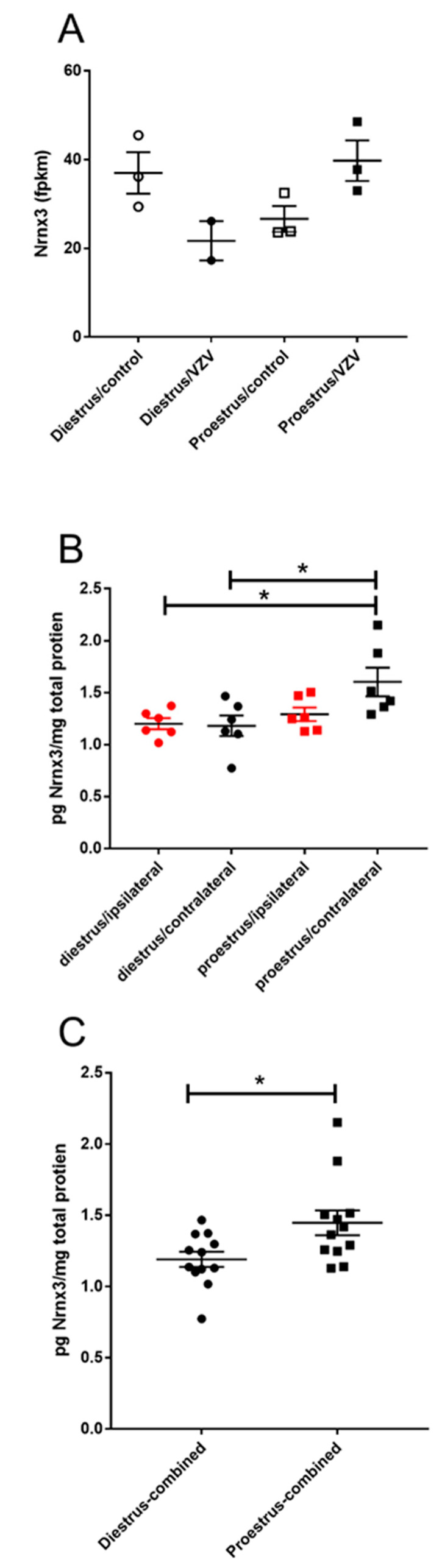
Expression of neurexin 3 (Nrxn3) significantly increased in the parabrachial region during proestrus. Rats were injected with varicella zoster virus in the left whisker pad to induce orofacial pain. Parabrachial brain tissue was collected from these female rats during the proestrus and diestrus phase of the estrous cycle. This tissue was analyzed for RNA (panel **A**) and protein (panels **B**,**C**). In panel (**A**) control animals received an injection of 0.9% saline in the left whisker pad. In panel (**B**) the right (contralateral to VZV injection) and left (ipsilateral to VZV injection) side of the brain was analyzed separately and in panel (**C**) the protein content from both sides of the brain were combined. Fragments Per Kilobase of transcript per Million mapped reads (fpkm). The asterisk indicates a significant difference between groups *p* < 0.05.

**Figure 3 ijms-21-05749-f003:**
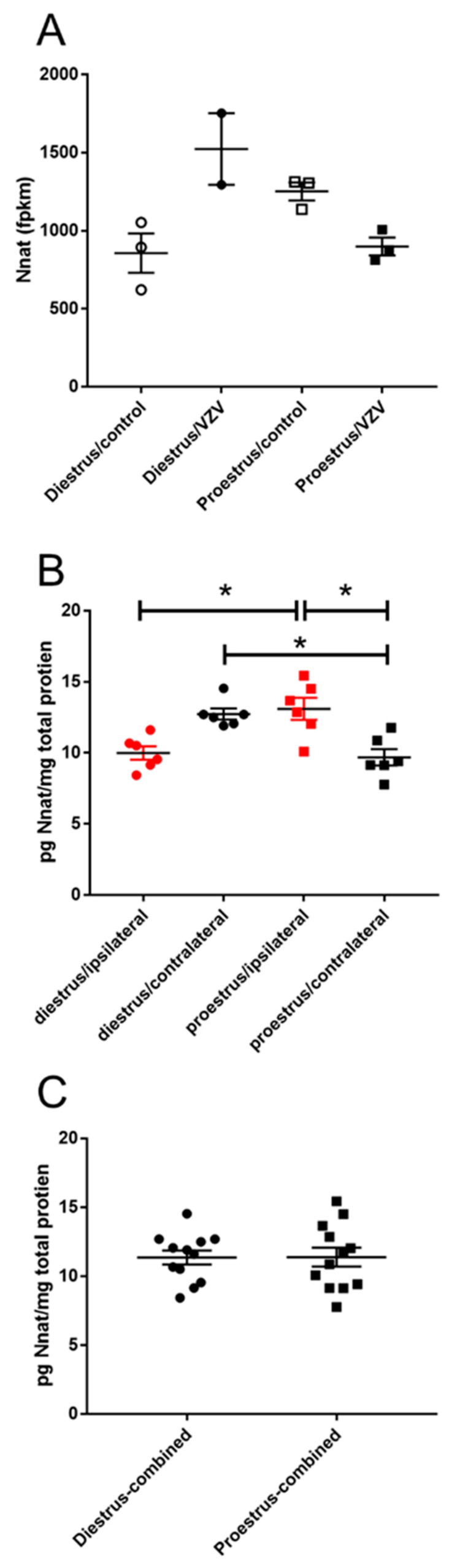
Expression of neuronatin (Nnat) significantly increased in the parabrachial region during diestrus. Rats were injected with varicella zoster virus in the left whisker pad to induce orofacial pain. Parabrachial brain tissue was collected from these female rats during the proestrus and diestrus phase of the estrous cycle. This tissue was analyzed for RNA (panel **A**) and protein (panels **B**,**C**). In panel (**B**) the right (contralateral to VZV injection) and left (ipsilateral to VZV injection) side of the brain was analyzed separately and in panel (**C**) the protein content from both sides of the brain were combined. Fragments Per Kilobase of transcript per Million mapped reads (fpkm). The asterisk indicates a significant difference between groups *p* < 0.05.

**Figure 4 ijms-21-05749-f004:**
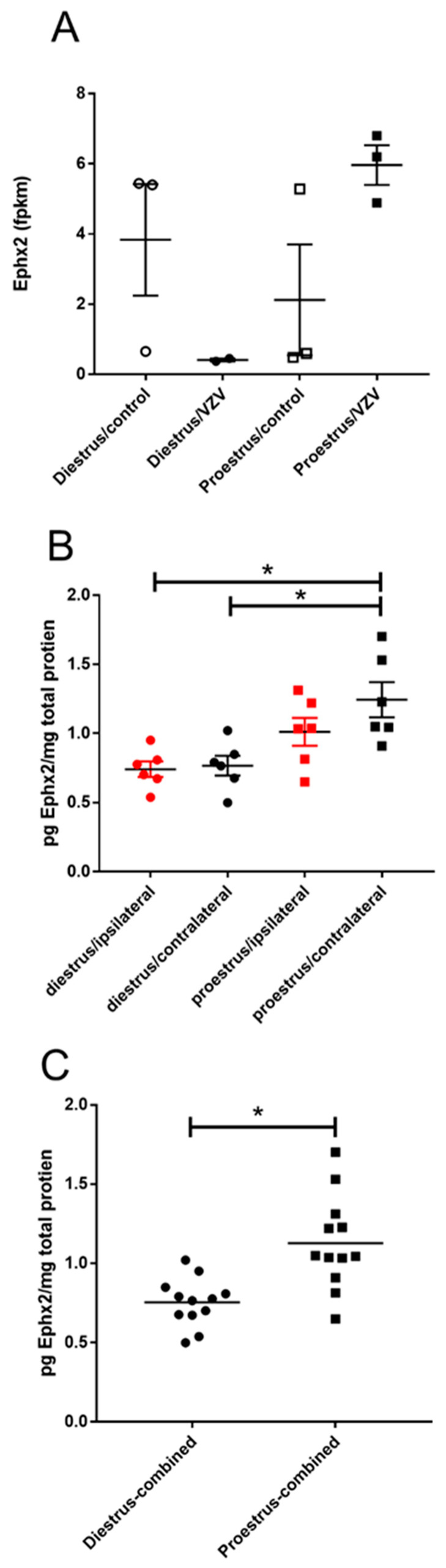
Expression of epoxide hydroxylase 2 (Ephx2) significantly increased in the parabrachial region during proestrus. Rats were injected with varicella zoster virus in the left whisker pad to induce orofacial pain. Parabrachial brain tissue was collected from these female rats during the proestrus and diestrus phase of the estrous cycle. This tissue was analyzed for RNA (panel **A**) and protein (panels **B**,**C**). In panel (**B**) the right (contralateral to VZV injection) and left (ipsilateral to VZV injection) side of the brain was analyzed separately and in panel (**C**) the protein content from both sides of the brain were combined. Fragments Per Kilobase of transcript per Million mapped reads (fpkm). The asterisk indicates a significant difference between groups *p* < 0.05.

**Figure 5 ijms-21-05749-f005:**
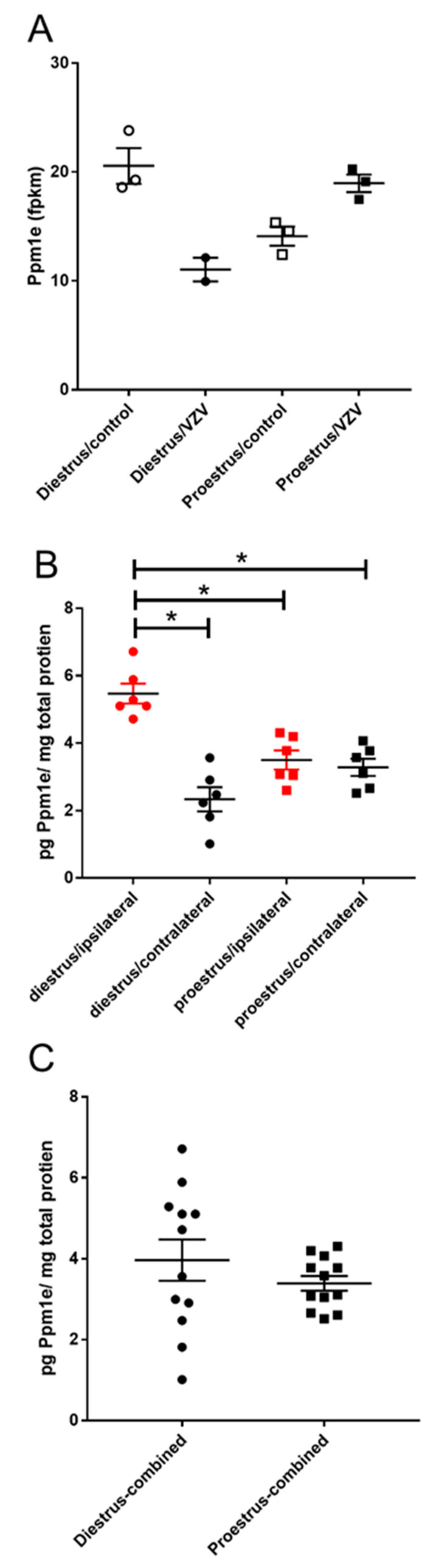
Expression of protein phosphatase, Mg^2+^/Mn^2+^ dependent 1E (Ppm1e) significantly increased in the parabrachial region at diestrus. Rats were injected with varicella zoster virus in the left whisker pad to induce orofacial pain. Parabrachial brain tissue was collected from these female rats during the proestrus and diestrus phase of the estrous cycle. This tissue was analyzed for RNA (panel **A**) and protein (panels **B**,**C**). In panel (**B**) the right (contralateral to VZV injection) and left (ipsilateral to VZV injection) side of the brain was analyzed separately and in panel (**C**) the protein content from both sides of the brain were combined. Fragments Per Kilobase of transcript per Million mapped reads (fpkm). The asterisk indicates a significant difference between groups *p* < 0.05.

**Figure 6 ijms-21-05749-f006:**
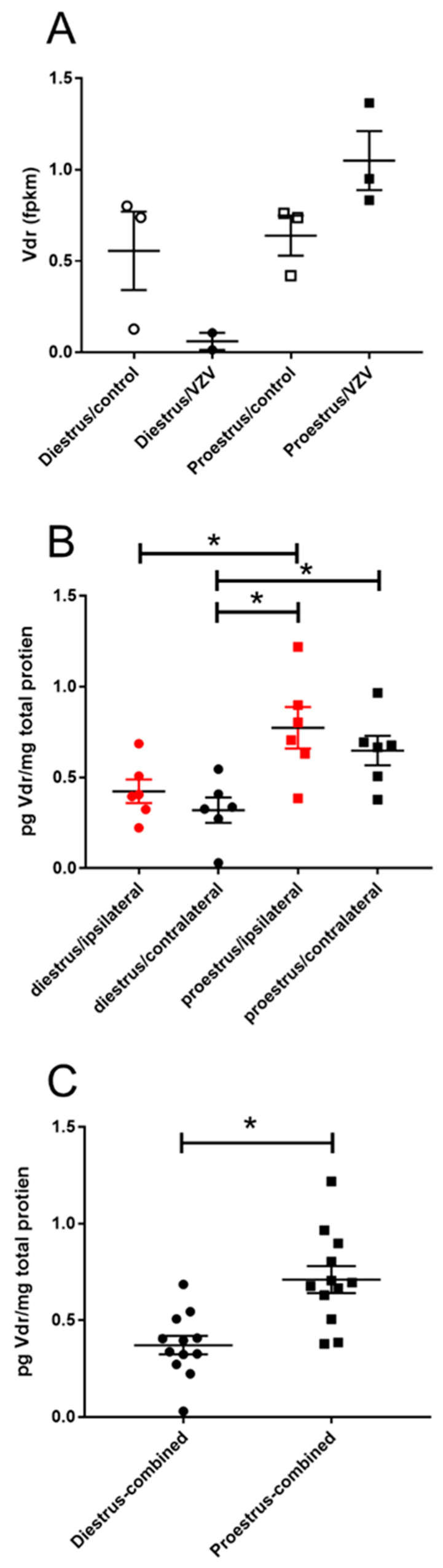
Expression of vitamin D receptor (Vdr) significantly increased in the amygdala region during proestrus. Rats were injected with varicella zoster virus in the left whisker pad to induce orofacial pain. Central amygdala brain tissue was collected from these female rats during the proestrus and diestrus phase of the estrous cycle. This tissue was analyzed for RNA (panel **A**) and protein (panels **B**,**C**). In panel (**B**) the right (contralateral to VZV injection) and left (ipsilateral to VZV injection) side of the brain was analyzed separately and in panel (**C**) the protein content from both sides of the brain were combined. Fragments Per Kilobase of transcript per Million mapped reads (fpkm). The asterisk indicates a significant difference between groups *p* < 0.05.

**Figure 7 ijms-21-05749-f007:**
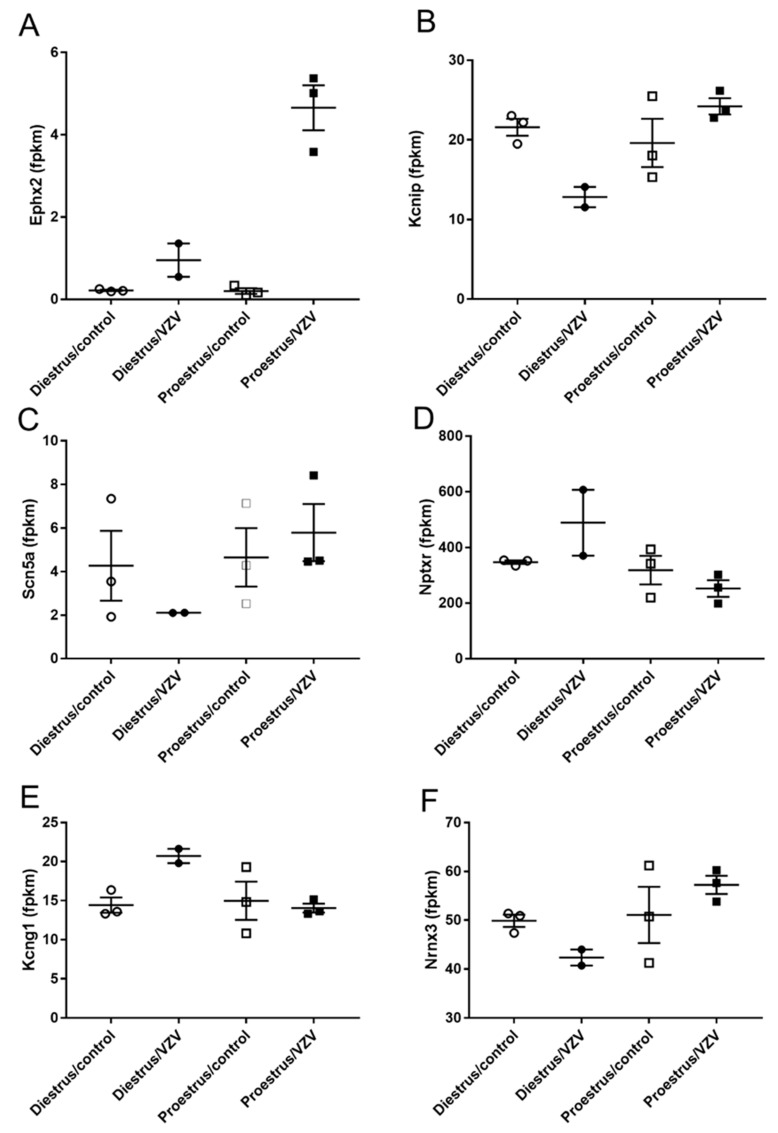
Expression of transcript for genes functioning in pain within the central amygdala region after VZV injection. Rats were injected with varicella zoster virus or vehicle in the left whisker pad to induce orofacial pain. Central amygdala brain tissue was collected from these female rats during the proestrus and diestrus phase of the estrous cycle. RNA-seq analysis for the whole transcriptome was completed for Ephx2 (panel **A**), Kcnip1 (panel **B**), Scn5a (panel **C**), Nptxr (panel **D**), Kcng1 (panel **E**) and Nrnx3 (panel **F**). Fragments Per Kilobase of transcript per Million mapped reads (fpkm).

**Table 1 ijms-21-05749-t001:** Rat estrous phase information for samples used in RNA-seq.

Rat ID	Vaginal Smear Phase of Estrous	Plasma Estradiol pg/mL
control diestrus rat #1	Diestrus	27
control diestrus rat #2	Diestrus	24
control diestrus rat #3	Diestrus	20
control proestrus rat #4	Proestrus	55
control proestrus rat #5	Proestrus	48
control proestrus rat #6	Proestrus	49
VZV diestrus rat #1	Diestrus	32
VZV diestrus rat #2	Diestrus	35
VZV diestrus rat #3	Diestrus	42
VZV proestrus rat #4	Proestrus	58
VZV proestrus rat #5	Proestrus	78
VZV proestrus rat #6	Proestrus	55

**Table 2 ijms-21-05749-t002:** RNA-seq gene expression in parabrachial after injection of control or varicella zoster virus.

**Gene Name**	***p* Value**	**Control Diestrus Rat #1 (fpkm)**	**Control Diestrus Rat #2 (fpkm)**	**Control Proestrus Rat #3 (fpkm)**	**Control Proestrus Rat #4 (fpkm)**	**Control Proestrus Rat #5 (fpkm)**	**Control Proestrus Rat #6 (fpkm)**
*Ephx2*	0.53508	5.401536737	0.652275294	5.440173033	0.485345846	0.595509868	5.283702722
*Ppm1e*	0.001275	18.6049103	23.82209521	19.26962088	14.57436205	12.41609143	15.34485813
*Nnat*	0.000492	1052.72778	621.1413696	894.5597825	1305.664627	1136.928297	1312.763391
*Nrxn3*	0.020639	29.37733025	45.50262412	36.15952131	23.84571693	23.58519927	32.46878525
**Gene Name**	***p* Value**	**VZV Diestrus Rat #1 (fpkm)**	**VZV Diestrus Rat #2 (fpkm)**		**VZV Proestrus Rat #4 (fpkm)**	**VZV Proestrus Rat #5 (fpkm)**	**VZV Proestrus Rat #6 (fpkm)**
*Ephx2*	5.50 × 10^−21^	0.371007388	0.45080559		6.200955306	4.887088102	6.801894656
*Ppm1e*	0.000565	12.13319367	9.953304244		19.12443259	20.27487947	17.49448297
*Nnat*	0.000767	1294.844022	1753.18802		875.8921637	1007.340802	812.1635673
*Nrxn3*	0.003501	26.12487589	17.2828174		48.53559573	33.00763395	37.77071015

**Table 3 ijms-21-05749-t003:** RNA-seq gene expression in central amygdala after infection with saline or varicella zoster virus in cycling female rats.

**Gene Name**	***p* Value**	**Control Diestrus Rat #1 (fpkm)**	**Control Diestrus Rat #2 (fpkm)**	**Control Diestrus Rat #3 (fpkm)**	**Control Proestrus Rat #4 (fpkm)**	**Control Proestrus Rat #5 (fpkm)**	**Control Proestrus Rat #6 (fpkm)**
*Vdr*	0.76741	0.128463801	0.801421678	0.739376681	0.760956766	0.737619776	0.419872491
*Ephx2*	0.38483	5.144149024	0.331596021	5.401425329	0.329489306	0.663337126	3.964683044
*Kcnip1*	0.32665	19.50941247	23.0344028	22.20121629	18.03663967	25.4747876	15.31723787
*Scn5a*	0.91535	1.925203264	7.339465076	3.545136417	2.53303478	7.126006971	4.28974909
*Nptxr*	0.28919	353.8862751	334.440786	352.5947033	219.7968486	341.6320859	393.3979149
*Kcng1*	0.9709	13.31573766	13.60014634	16.36389723	10.80707018	19.29723049	14.81903359
*Nrxn3*	0.87119	50.90993298	51.38604069	47.36862436	41.2493791	61.21788973	50.74334292
**Gene Name**	***p* Value**	**VZV Diestrus Rat #1 (fpkm)**	**VZV Diestrus Rat #2 (fpkm)**		**VZV Proestrus Rat #4 (fpkm)**	**VZV Proestrus Rat #5 (fpkm)**	**VZV Proestrus Rat #6 (fpkm)**
*Vdr*	2.15 × 10^−7^	0.014367	0.107072963		1.36584192	0.950491632	0.8329427
*Ephx2*	5.23 × 10^−7^	1.356616613	0.545643822		5.373784519	5.013868211	3.581573442
*Kcnip1*	1.45 × 10^−6^	11.54816524	14.09685865		26.16338119	22.77739586	23.7018345
*Scn5a*	9.89 × 10^−6^	2.112411356	2.10566044		8.408643753	4.462007618	4.5033157
*Nptxr*	8.44 × 10^−5^	606.5933969	370.7522157		198.5722507	302.1822918	256.01793
*Kcng1*	7.82 × 10^−5^	21.62632637	19.7959817		13.32858529	13.6335783	15.166255
*Nrxn3*	0.000433	40.71278489	43.98085179		60.25477266	57.65077434	53.820079

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
