# Peer review of "Comparing Gene Expression in the Parabrachial and Amygdala of Diestrus and Proestrus Female Rats after Orofacial Varicella Zoster Injection"

_ijms, 2020, doi:10.3390/ijms21165749_

Round 1

Reviewer 1 Report

This manuscript by Hornung et al. attempts to identify neuronal genes involved and mechanisms of sex steroid hormone responsive orofacial pain. They used a female rat model of viral induced orofacial pain by injecting varicella zoster virus (VZV) into the whisker pads and characterizing changes in gene expression in two areas of the brain, parabrachial and amygdala, during low estradiol (diestrus) and high estradiol (proestrus). I have several suggested revisions for improving the manuscript, described below, with additional explanation or presentation needed to better understand the manuscript.

  1. In the Intro, the relevance and significance of sex steroid effects on post herpetic neuralgia pain in people is not described and needs to be included.
  2. With the multiple groupings there were only 2 or 3 animals per sub-group. This must be addressed for significance of findings.
  3. The plasma estradiol levels for diestrus and Proestrus are not that different. What is the length of the cycle? What range of estradiol levels were found throughout the animal cycle and was there much variation?
  4. The timing of diestrus and proestrus in the rat model needs to be defined. The authors could consider adding a graph showing the levels found throughout the cycle with the diestrus and proesterus times identified.
  5. Table 1 Lists the Diestrus data before the Proestrus data and should switch order. Explain why they are not that different.
  6. Table 2 is very busy and confusing. The description given to identify the genes to be analyzed is very hard to follow. Please clarify. Some of the highlighted genes do not show much difference between diestrus and proestrus. Also please explain p values of 8.14E-05 to 4.28E-79?
  7. Figures 1-6. The range of gene expression for each of the genes is very different. Please address in the results and discussion.

Author Response

Reviewer #1

Comment #1: In the Intro, the relevance and significance of sex steroid effects on post herpetic neuralgia pain in people is not described and needs to be included.

Reply #1: To the second paragraph of the introduction these statements have been added, “Varicella zoster virus (VZV) infection results in viral latency [19-22] and often VZV re-activates resulting in what is termed herpes zoster (HZ). Unfortunately HZ causes pain in up to 90% patients [23].  This zoster-associated pain occurs before or near the time skin lesions are observed and post-herpetic neuralgia, (PHN) is pain defined to begin 3 months after the resolution of the lesions [24-26].  VZV injected into the footpad of either Wistar or Sprague Dawley rats results in measurable hypersensitivity that is similar to that in humans with PHN [27-33].  Treatment of these animals with antivirals, opioids or NSAIDS mirrors PHN patients treated with these same drugs, in that, zoster associated pain does not respond to antivirals, but does respond to treatment with gabapentin or sodium channel blockers [29, 30, 33, 34].  In models of post-herpetic neuralgia sex steroids alter the orofacial pain response [35]. Interestingly, females report PHN 3.75 times more often than men such that this pain has been reported to be 36% higher in females [36, 37].  Thus, altered gene expression due to changes in sex steroids would be expected in this model.” 

Comment #2: With the multiple groupings there were only 2 or 3 animals per sub-group. This must be addressed for significance of findings.

Reply #2: To the statistics section in the Methods we have added the statement, “Note: RNA-seq data included 2-3 animals per treatment group, power analysis calculations for a level 0.05 tests using the mean and standard deviation values indicated a range of 30% power for the Vdr gene to over 90% power for the Ppm1e and Ephx2 gene.  A standard of 80% power is acceptable but we are below that standard for Nrnx3, Nnat, Scn5a, Vdr and Nptxr.”

Comment #3: The plasma estradiol levels for diestrus and Proestrus are not that different. What is the length of the cycle? What range of estradiol levels were found throughout the animal cycle and was there much variation?

Reply #3: The length of the cycle will vary around 4 to 5 days.  We have added a new figure, Figure 1, that shows estradiol for a typical 5 day estrous cycle, panel A.  Periods in which tissue was collected are added to Figure 1A.  The estradiol concentration does not change much, other than proestrus, panel A.

Comment #4: The timing of diestrus and proestrus in the rat model needs to be defined. The authors could consider adding a graph showing the levels found throughout the cycle with the diestrus and proesterus times identified.

            Reply #4:  We have added a graph of plasma estradiol for intact female rats over a five day period (Figure 1A).  On this graph the collection periods for harvesting brain tissue are indicated.  Brain tissue was harvested based on one, the vaginal smear cytology (Figure 1B and Figure 1C) and two, the estradiol levels (Figure 1A).

Comment #5: Table 1 Lists the Diestrus data before the Proestrus data and should switch order. Explain why they are not that different.

Reply#5: We have redone Table 1 to list the diestrus data before the proestrus data.  In the second paragraph of the methods “Treatment Groups” we have added details of how the stage of the estrous cycle was determined, it states, “Because the concentration of plasma estradiol varies over the estrous cycle (Fig. 1A) the differences between diestrus and proestrus should not be the only indicator and the vaginal smears aid in a positive determination (Fig. 1B, C and D).”

Comment #6: Table 2 is very busy and confusing. The description given to identify the genes to be analyzed is very hard to follow. Please clarify. Some of the highlighted genes do not show much difference between diestrus and proestrus. Also please explain p values of 8.14E-05 to 4.28E-79?

Reply #6: Table 2 has been simplified. We clarified the description for the genes to be analyzed on page 9 of the results section to state, “Genes to be analyzed at the protein level were selected on two criteria first, the genes that had the greatest change in fpkm between diestrus and proestrus were analyzed (pvalue of <0.005) and second, the gene had been reported to function in pain.”

The pvalue is a measure of how likely you are to get this data if no real difference existed (i.e., got significant data but it really was not significant). Therefore, a small pvalue indicates that there is a small chance of getting this data if no real difference existed.  In other words, a pvalue = 0.05 means that there is a 5% chance for a not-differentially expressed gene to show these kind of expression differences. But: with 10,000 genes (i.e. 10,000 tests) you can expect 0.05 x 10,000 = 500 false positives!  To correct for this potential error in the pvalue for doing a large number of tests an adjusted p-value (padj) is calculated.  padj = 0.05 means that there is a 5% chance that these expression values are from a not differentially expressed gene, see supplementary data.

A pvalue of 8.14E-0.05 would mean that there is a 0.008 % chance that a gene having no real difference would randomly (or by chance due to variation) show a difference. 

Comment #7: Figures 1-6. The range of gene expression for each of the genes is very different. Please address in the results and discussion.

Reply #7: To the discussion on page 13 we added, “Note that the level of expression for Vdr was several fold lower than most genes.   For example, the Vdr fkpm ranged from 0.01 to 1.3 in the amygdala but Nrxn3 ranged from 40 to 61.  When looking at protein expression, Vdr ranged between 0.4 and 0.7 pg/mg but Nnat was between 10 and 13 pg/mg (mean values).  Thus, not only were there differences between treatment groups there were dramatic differences between the level of transcript between different genes.”

Reviewer 2 Report

ijms-866775

Brief summary

The study by Hornung et al. entitled ‘ Comparing gene expression in the parabrachial and amygdala of diestrus and proestrus female rats after orofacial varicella zoster injection’ showed that several genes that have a known role in pain signaling displayed altered expression after VZV injection in either the parabrachial or amygdala regions, in diestrus and proestrus in rats.

The article contains many interesting pieces of new information that could be of real interest to the scientific community. The results are based on RNA-Seq and ELISA analysis. This multi-approach analysis works with RNA and protein expression as well. The design of the study, however, shows many minor issues and the text, tables, and figures also need to be corrected and upgraded to increase the quality of the manuscript and send a much clearer scientific message. 

I will list these issues in my specific comments.

After answering and correcting these I see no obstacles ahead to publish this paper in IJMS.

Specific comments

Major comments

  1. The most disturbing is the misleading description of controls, vehicle, and VZV-titers. ‘100 µl of MeWo cells infected with VZV (50,000-100,000 pfu/µl) or vehicle (0.9% saline). VZV had MeWo (human melanoma derived cell line) as ‘vehicle’, but 0.9% saline had been used as vehicle. VZV titers could have a 2-fold difference, that could substantially affect the results. 
  2. In relation to my previous comment: Could MeWo cells cause inflammation and pain, mimicking the effects of the VZV? In the present form, the article sends an unclear message, as it only deals with VZV-induced pain and not mentioning the possibility of the RNA/protein expression changes as a result of viral and/or ‘real vehicle’ injections.
  3. In the introduction, the authors do not mention any papers dealing with VZV injection as a model for pain. Please, provide some background (e.g. Kinchington 2011; Guedon 2014).
  4. My other concern is the quality and clarity of tables and figures. The VZV+ and Control animals should show up next to each other as a comparison from the different regions as well. Please, indicate RNA gene name on the figure with FPKM, instead of writing it in the long form.
  5. Table 1. The cycle would be more understandable with representative images of estrus cycle smears.
  6. Table 2. and 3. are overcrowded and go far beyond the margin of the page. The data presented here would be clearer in a color-coded figure with the gene expression delta-fpkm showed.
  7. What is NEWGENE_1306267 in Table 2? (In the NCBI Nucleotide DB it’s Rattus norvegicus phosphopantothenoylcysteine decarboxylase.)
  8. Fig 1. The quality of fig1. is low. The image is pixelated and on B, C Y-axis please change protien ->protein. ‘Rats were…’-on all figure legends.
    In Fig2-5., the same comments as in Fig1.
  9. The text contains many spelling mistakes and typos that could have been avoided by using a spell checker on the manuscript. I tried to find as many as possible to help the work of the authors, but it needs a major re-check before the final version (see minor comments).

Minor comments

Abstract

L12. effect->affect

L13. effect->affect

L14. ‘the’ expression

L15. effect->affect

Introduction

L41. Currently, 

L43. neuralgia,

L50. were selected for quantitation of protein content in the parabrachial and amygdala-tissues by ELISA.

Methods

L62. a unilateral

L64.  diestrus/vzv -> VZV

L69. isolated 2x

L96. Paired-end

L105. protein

L108. protein

L115. Please, clarify the sentence.

L117. data is available

Results

L120. were identified

L131. second, the

L133. studies, we

L143. Rats were injected

L158. Rats were

L163. The asterisk

L215., 219. in the transcript

Figure 6. Rats were injected…

Discussion

L228. include

L237. The deletion

L242. affect

L244. affect

L244. Nnat overexpression 

L250. In this study, 

L265. affects

L269. amygdala, only

L280. or amygdala

Supp.

L286. the vehicle

L288. the vehicle

Author Response

Reviewer #2

Major comments

Comment #1: The most disturbing is the misleading description of controls, vehicle, and VZV-titers. ‘100 µl of MeWo cells infected with VZV (50,000-100,000 pfu/µl) or vehicle (0.9% saline). VZV had MeWo (human melanoma derived cell line) as ‘vehicle’, but 0.9% saline had been used as vehicle. VZV titers could have a 2-fold difference, that could substantially affect the results. 

            Reply #1: In the first paragraph of the discussion on page 11 we address this issue, “A saline control was performed to contrast changes in gene expression due to MeWo cell and virus injection although MeWo cells without virus are the typical control to determine the effects of VZV [27, 30, 33].”

We also clarify that the different titers were included in each group to minimize the effect of titer on the results, in the methods on page 6 we now state, “The viral titers were distributed so that 1-2 animals in each group were injected with the lower titer cells (i.e., 50,000 pfu/µl) and 1-2 animals were injected with the higher titer cells (i.e., 100,000 pfu/µl).”

Comment #2: In relation to my previous comment: Could MeWo cells cause inflammation and pain, mimicking the effects of the VZV? In the present form, the article sends an unclear message, as it only deals with VZV-induced pain and not mentioning the possibility of the RNA/protein expression changes as a result of viral and/or ‘real vehicle’ injections.

Reply #2: Yes, MeWo cells alone could cause inflammation but in our previous work we have not seen an increase in pain 7 days post injection when comparing a PBS injected group and animals receiving injection of MeWo cells without virus (Stinson et al. BMC Neurology (2017) 17:95).  Thus, it is unlikely the change in gene expression (pain pathway genes) are the result of MeWo cells.

Comment #3: In the introduction, the authors do not mention any papers dealing with VZV injection as a model for pain. Please, provide some background (e.g. Kinchington 2011; Guedon 2014).

Reply #3: Reviewer #1 also had this same comment. To the introduction these statements have been added, “Varicella zoster virus (VZV) infection results in viral latency [19-22] and often VZV re-activates resulting in what is termed herpes zoster (HZ). Unfortunately HZ causes pain in up to 90% patients [23].  This zoster-associated pain occurs before or near the time skin lesions are observed and post-herpetic neuralgia, (PHN) is pain defined to begin 3 months after the resolution of the lesions [24-26].  VZV injected into the footpad of either Wistar or Sprague Dawley rats results in measurable hypersensitivity that is similar to that in humans with PHN [27-33].  Treatment of these animals with antivirals, opioids or NSAIDS mirrors PHN patients treated with these same drugs, in that, zoster associated pain does not respond to antivirals, but does respond to treatment with gabapentin or sodium channel blockers [29, 30, 33, 34].  In models of post-herpetic neuralgia sex steroids alter the orofacial pain response [35]. Interestingly, females report PHN 3.75 times more often than men such that this pain has been reported to be 36% higher in females [36, 37].  Thus, altered gene expression due to changes in sex steroids would be expected in this model.” 

Comment #4: My other concern is the quality and clarity of tables and figures. The VZV+ and Control animals should show up next to each other as a comparison from the different regions as well. Please, indicate RNA gene name on the figure with FPKM, instead of writing it in the long form.

Reply #4: The figures have been modified and the gene name is present on the graphs.  Each treatment group is now a different shaped symbol or the symbol is colored.  At the protein level, no single gene changed significantly in both the parabrachial and amygdala.  A figure comparing the parabrachial and amygdala can be added if requested.

Comment #5: Table 1. The cycle would be more understandable with representative images of estrus cycle smears.

Reply #5: This is similar to a comment by reviewer #1.  We have added a graph of plasma estradiol for intact female rats over a five day period (Figure 1A).  On this graph the collection periods for harvesting brain tissue are indicated.  Brain tissue was harvested based on one, the vaginal smear cytology (Figure 1B and Figure 1C) and two, the estradiol levels (Figure 1A).

Comment #6: Table 2. and 3. are overcrowded and go far beyond the margin of the page. The data presented here would be clearer in a color-coded figure with the gene expression delta-fpkm showed.

Reply #6: Table 2 and 3 have been simplified to include only those genes analyzed in the figures and the tables have been expanded to include the control data. 

Comment #7: What is NEWGENE_1306267 in Table 2? (In the NCBI Nucleotide DB it’s Rattus norvegicus phosphopantothenoylcysteine decarboxylase.)

Reply #7: This information has been eliminated from Table 2 to simplify the data.  The gene is still listed as NEWGENE_1306267 in the supplementary data (how we received the data) but can be renamed at reviewer’s request.

Comment #8: Fig 1. The quality of fig1. is low. The image is pixelated and on B, C Y-axis please change protien ->protein. ‘Rats were…’-on all figure legends.
In Fig2-5., the same comments as in Fig1.

Reply #8: All figures have been redone and uploaded.  Legends have been corrected.

Comment #9: The text contains many spelling mistakes and typos that could have been avoided by using a spell checker on the manuscript. I tried to find as many as possible to help the work of the authors, but it needs a major re-check before the final version (see minor comments).

Reply #9: These edits have been incorporated into the manuscript.

Minor comments

Reply: These edits have been incorporated into the manuscript.

Abstract

L12. effect->affect

L13. effect->affect

L14. ‘the’ expression

L15. effect->affect

Introduction

L41. Currently, 

L43. neuralgia,

L50. were selected for quantitation of protein content in the parabrachial and amygdala-tissues by ELISA.

Methods

L62. a unilateral

L64.  diestrus/vzv -> VZV

L69. isolated 2x

L96. Paired-end

L105. protein

L108. protein

L115. Please, clarify the sentence.

L117. data is available

Results

L120. were identified

L131. second, the

L133. studies, we

L143. Rats were injected

L158. Rats were

L163. The asterisk

L215., 219. in the transcript

Figure 6. Rats were injected…

Discussion

L228. include

L237. The deletion

L242. affect

L244. affect

L244. Nnat overexpression 

L250. In this study, 

L265. affects

L269. amygdala, only

L280. or amygdala

Supp.

L286. the vehicle

L288. the vehicle

Round 2

Reviewer 1 Report

The reviewer appreciates how thoroughly the comments were addressed. Those changes resulted in a much improved, clear and very compelling manuscript that is approved for publication.